# Memory Layers at Scale

**Vincent-Pierre Berges** [* 1] **Barlas Oğuz** [* 1] **Daniel Haziza** [1] **Wen-tau Yih** [1] **Luke Zettlemoyer** [1] **Gargi Ghosh** [1]

## Abstract

Memory layers use a trainable key-value lookup mechanism to add extra parameters to a model without increasing FLOPs. Conceptually, sparsely activated memory layers complement compute-heavy dense feed-forward layers, providing dedicated capacity to store and retrieve information cheaply. This work takes memory layers beyond proof-of-concept, proving their utility at contemporary scale. On downstream tasks, language models augmented with our improved memory layer outperform dense models with more than twice the computation budget, as well as mixture-of-expert models when matched for both compute and parameters. We find gains are especially pronounced for factual tasks. We provide a fully parallelizable memory layer implementation, demonstrating scaling laws with up to 128B memory parameters, pretrained to 1 trillion tokens, comparing to base models with up to 8B parameters.

## 1. Introduction

Pretrained language models encode vast amounts of information in their parameters (Roberts et al., 2020), and they can recall and use this information more accurately with increasing scale (Brown et al., 2020). For dense deep neural networks, which encode information primarily as weights of linear matrix transforms, this scaling of parameter size is directly coupled to an increase in computational and energy requirements. It is unclear if this is the most efficient solution to all information storage needs of language models. An important subset of information that language models need to learn are simple associations. For example, LLMs learn birthdays of celebrities, capital cities of countries, or how one concept might relate to another. While feed-forward networks can in principle (given sufficient scale)

---

[*]Equal contribution [1]Meta FAIR. Correspondence to: Vincent-Pierre Berges <vincentpierre@meta.com>, Barlas Oğuz <barlaso@meta.com>.

*Proceedings of the 42nd International Conference on Machine Learning*, Vancouver, Canada. PMLR 267, 2025. Copyright 2025 by the author(s).

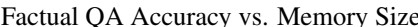

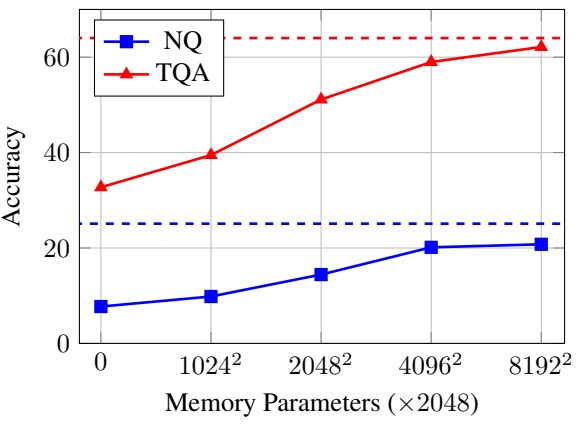

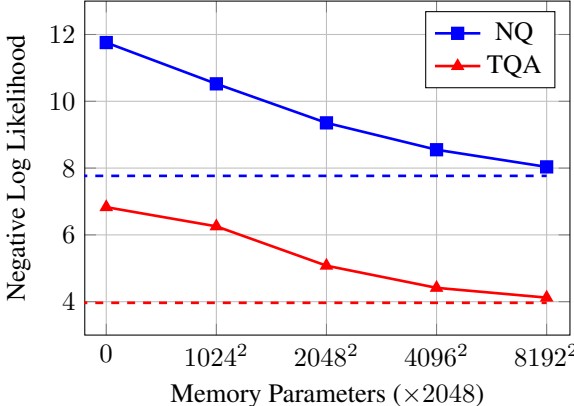

*Figure 1.* Scaling the size of the memory for a 1.3 billion parameter base model (zero memory parameters corresponds to a dense model), trained to 1 trillion tokens. On the left, factual QA accuracy (exact match on NaturalQuestions and F1 score on TriviaQA), on the right task NLL (lower is better). Dashed lines show the performance of a 7B model trained on 2 trillion tokens with 10x more FLOPs.

learn any function (Hornik et al., 1989), including lookup tables of associations, using an associative memory for this purpose would be both more efficient and more natural.

Such memory layers can be implemented with a simple and cheap key-value lookup mechanism where both keys and values are encoded as embeddings (Weston et al., 2015).

Earlier works introduced end-to-end trainable memory layers (Sukhbaatar et al., 2015) and incorporated them as part of neural computational systems (Graves et al., 2014). Despite early enthusiasm however, memory layers have not been studied and scaled sufficiently to be useful in modern AI architectures. There are distinctive challenges one encounters when attempting to scale memory layers, which we touch upon in Section 3. In contrast to dense layers which are predominantly FLOP-bound, memory layers with their sparse activation pattern are almost entirely memory bandwidth bound. Such components are rarely used in modern architectures and have not been optimised for hardware accelerators. In addition to, and partly as a result of this, little research was done to improve their performance. Instead, the field focused on alternatives such as mixture-of-experts (Shazeer et al., 2017), which more closely resemble dense networks and are thus easier to scale, but are challenging to optimize for inference.

In this work, we show that memory layers, when improved and scaled sufficiently, can be used to augment dense neural networks to great benefit. We do so by replacing the feed-forward network (FFN) of one or more transformer layers with memory layers (we leave other layers unchanged). These benefits are consistent across a range of base model sizes (ranging from 134 million to 8 billion parameters), and memory capacities (up to 128 billion parameters). This represents a two orders of magnitude leap in memory capacity compared to previous memory layers reported in the literature. Our results (Section 4) indicate that memory layers improve the factual accuracy of language models by over 100% as measured by factual QA benchmarks, while also improving significantly on coding (HumanEval, MBPP) and general knowledge (Hellaswag, MMLU). In many cases, memory augmented models can match the performance of dense models that have been trained on 4x more compute. They also outperform mixture-of-experts architectures with matching compute and parameter size, especially on factual tasks. Given these findings, we strongly advocate that memory layers be integrated into all next generation AI architectures. [1]

## 2. Related work

Language model scaling laws (Kaplan et al., 2020) study the empirical performance of language models as they are scaled in compute, data, and parameter size. Scaling laws are typically formulated in terms of training/test log likelihood, which is generally believed to correlate well with downstream performance. Scaling plots on downstream tasks are also not without precedent (Brown et al., 2020), but have sometimes been shown to exhibit non-linear be-

haviour and phase transitions (Wei et al., 2022; Ganguli et al., 2022). Nevertheless, given a well behaved metric (such as task likelihood loss), most tasks exhibit smooth improvements with scaling (Schaeffer et al., 2023).

(Kaplan et al., 2020) showed that performance scales log-linearly with compute and parameter size across a wide range of architecture hyper-parameters, such as model depth and width. It has been difficult to find architectures which substantially deviate from these laws. Mixture-of-experts (MOE) (Shazeer et al., 2017; Lepikhin et al., 2020) is a notable exception. MOE adds extra parameters to the model without increasing the computation budget. While scaling laws for MOE also mostly focus on training perplexity, gains transfer well to downstream applications, as evidenced by the popularity of MOE architectures in recent state-of-the-art model families (Jiang et al., 2024; OpenAI et al., 2024; Team et al., 2024). Nevertheless, scaling laws for specific task families and capabilities like factuality remain understudied.

Like MOE, memory augmented models also aim to augment the parameter space of the model without adding significant computational cost. Memory networks were proposed initially in (Weston et al., 2015), and with end-to-end training in (Sukhbaatar et al., 2015). Neural Turing Machines (Graves et al., 2014; 2016) combine external trainable memory with other components to build a neural trainable computer. Product-key networks (Lample et al., 2019) were introduced to make the memory lookup more efficient and scalable. The recent PEER (He, 2024) builds on this work, replacing vector values with rank-one matrices, forming a bridge between memory architectures and MOE. Memory networks make use of a form of sparse attention using a top-k lookup mechanism similar to Top-k Attention (Gupta et al., 2021). Moreover, the idea of replacing MLPs with attention mechanisms is similar to what was done in "Augmenting Self-attention with Persistent Memory" (Sukhbaatar et al., 2019).

Factual text generation has long been considered a fundamental capability for generative models, typically benchmarked through factual open domain question answering (Chen et al., 2017; Chen & Yih, 2020) and other knowledge-intensive tasks (Petroni et al., 2021). Being able to memorize the facts in the training corpus enables the model to answer fact-seeking, knowledge intensive tasks more factually and accurately. Indeed larger models have been shown to be more factual (Roberts et al., 2020; Brown et al., 2020), but even modern LLMs are known to struggle with hallucination (Ji et al., 2023). A tested way of ensuring more factuality is through retrieval augmented generation (Lewis et al., 2021; Karpukhin et al., 2020; Lee et al., 2019; Guu et al., 2020; Khandelwal et al., 2020). We use short-form QA tasks in this work to demonstrate the effec-

---

[1]Our implementation is available at `https://github.com/facebookresearch/memory`

tiveness of memory layers and leave the long-form generation tasks for future work. Recently, a wide literature has emerged in mitigating LLM hallucinations through data related methods, architecture variants, pre-training and inference time improvements. We refer to (Ji et al., 2023) section 5 for a comprehensive survey.

# 3. Memory Augmented Architectures

Trainable memory layers work similarly to the ubiquitous attention mechanism (Bahdanau et al., 2016). Given a query $q \in \mathbb{R}^n$, a set of keys $K \in \mathbb{R}^{N \times n}$ and values $V \in \mathbb{R}^{N \times n}$, the output is a soft combination of values, weighted according to the similarity between $q$ and the corresponding keys. Two major differences separate memory layers from attention layers as they are typically used (Vaswani et al., 2023). First, the keys and values in memory layers are trainable parameters, as opposed to activations. Second, memory layers typically have larger scale in terms of the number of keys and values, making sparse lookup and updates a necessity. For example, in this work, we scale the number of key-value pairs to several millions. In this case, only the top-$k$ most similar keys and corresponding values take part in the output. A simple memory layer can be described by the following equations:

$$I = \text{TopkIndices}(Kq), \quad s = \text{Softmax}(K_I q), \quad y = sV_I \quad (1)$$

Here $I$ is a set of indices, $s \in \mathbb{R}^k$, $K_I, V_I \in \mathbb{R}^{k \times n}$, and the output $y \in \mathbb{R}^n$. Each token embedding (for us, the output of the previous attention layer) goes through this memory lookup independently, similar to the FFN operation that we replace.

## 3.1. Scaling memory layers

Being light on compute, and heavy on memory, memory layers have distinct scaling challenges. We detail some of these challenges and how we address them in this section.

### 3.1.1. PRODUCT-KEY LOOKUP

One bottleneck which arises when scaling memory layers is the query-key retrieval mechanism. A naive nearest-neighbour search requires comparing each query-key pair, which quickly becomes prohibitive for large memories. While fast approximate vector similarity techniques (Johnson et al., 2019) could be used here, it's a challenge to incorporate them when the keys are being continually trained and need to be re-indexed. Instead, we adopt trainable product-quantized keys from (Lample et al., 2019). Product keys work by having two sets of keys instead of one, where $K_1, K_2 \in \mathbb{R}^{\sqrt{N} \times \frac{n}{2}}$. The full set of keys of size $N \times n$, which is never instantiated, consists of the product of these two sets. The top-$k$ lookup on the full set of keys can be efficiently done by searching the much smaller set of half-keys first, saving compute and memory. To perform the lookup, we first split the query as $q_1, q_2 \in \mathbb{R}^{\frac{n}{2}}$.

Let $I_1, I_2$ and $s_1, s_2$ be the top-k indices and scores obtained from the respective key sets $K_1, K_2$. Since there are only $\sqrt{N}$ keys in each set, this operation is efficient. The overall indices and scores can be found by taking $\text{argmax}_{i_1 \in I_1, i_2 \in I_2} s_1[i_1] + s_2[i_2]$.

### 3.1.2. PARALLEL MEMORY

Memory layers are naturally memory-intensive, mostly due to the large number of trainable parameters and associated optimizer states. To implement them at the scale of several millions of keys, we parallelize the embedding lookup and aggregation across multiple GPUs. The memory values are sharded across the embedding dimension. At each step, the indices are gathered from the process group, each worker does a lookup and then aggregates the portion of embeddings in its own shard. After this, each worker gathers the partial embeddings corresponding to its own portion of the indices. We take care to keep activation memory manageable at this stage, by making sure each GPU only gets its own portion, and does not need to instantiate the entire embedding output. The process is illustrated in Figure 2. The implementation is independent of other model parallelism schemes such as tensor, context or pipeline parallelism, and operates on its own process group.

### 3.1.3. SHARED MEMORY

Deep networks encode information at different levels of abstraction across different layers. Adding memory to multiple layers may help the model use its memory in more versatile ways. In contrast to previous work (Lample et al., 2019), we use a shared pool of memory parameters across all memory layers, thus keeping parameter count the same and maximizing parameter sharing. We find that multiple memory layers increase performance significantly over having a single layer with the same total parameter count, up to a certain number of layers (in our case, 3). Beyond this point, replacing further FFN layers degrades performance, showing sparse and dense layers are both needed and likely complementary (see Section 5.4).

### 3.1.4. PERFORMANCE AND STABILITY IMPROVEMENTS

The main operation in the memory layer is to compute the weighted sum of the top-k embeddings: it is implemented in PyTorch's `EmbeddingBag` operation. As the number of floating-point operations is negligible, we expect this operation to be solely limited by the GPU memory bandwidth but find multiple inefficiencies in PyTorch's implementation in practice. We implemented new and more efficient CUDA kernels for this operation. Our forward pass optimizes memory accesses and achieves 3TB/s of memory bandwidth, which is close to our H100 specification of 3.35TB/s (compared to less than 400GB/s with PyTorch's

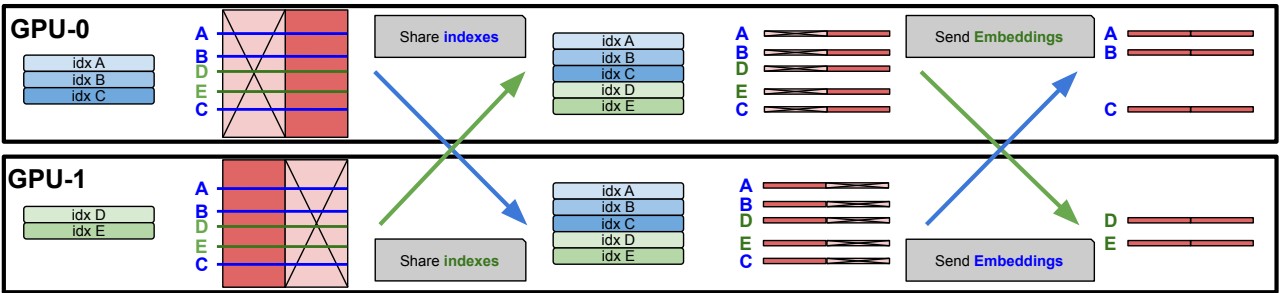

*Figure 2.* Illustration of the parallel EmbeddingBag implementation for a "Memory Group" of two GPUs. Each GPU performs the EmbeddingBag operation on all of the indices of the group, but on half-dimension embeddings it has access to.

implementation). The backward pass is more complicated as multiple output gradients have to be propagated to the same weight gradient. We benchmarked multiple strategies: using in-memory atomic additions, using locks, or computing the inverse mapping from embedding_id to token_ids to make the backward pass embarrassingly parallel. We exclusively used the atomic strategy in our trainings because it was the best performing, but the other ones were competitive, and could be used when the hardware does not support atomics in the given data type.

Overall, our custom kernels make the embedding bag operation end-to-end 6x faster compared to PyTorch's `EmbeddingBag` for our use cases.

### 3.2. Architectural improvements to memory layers

We improve training performance of the memory layer by introducing input-dependent gating with a `silu` non-linearity (Hendrycks & Gimpel, 2023). The output in Equation (1) then becomes

$$\text{output} = (y \odot \text{silu}(x^T W_1))^T W_2 \qquad (2)$$

where $\text{silu}(x) = x \, \text{sigmoid}(x)$ and $\odot$ is the element-wise multiplication(see also Figure 3). We find that for large memory layers, training can become unstable, especially for small base models. We use qk-normalization (Team, 2024) when needed to alleviate this issue.

### 3.3. Decoding inference efficiency

Memory layers have important advantages over alternatives like Mixture-of-Experts (MoE) when it comes to inference efficiency. Inference involves two phases: prefilling and decoding. Prefilling typically processes many tokens, and is in the same compute-bound regime as training, where memory layers only incur a small overhead. However, during decoding, a GPU typically processes only a few tokens at a time, due to either latency requirements, or memory limitations. This process is memory bandwidth bound: the decoding time for a single token depends on the size of

the activated parameters. Even with a small batch size, MoE models can activate a large portion of their parameters, making the decoding time many times slower than that of a dense model. Models with memory layers, on the contrary, only activate a small subset of the entire embedding table for every token in the batch, hence the number of parameters activated remains roughly constant in the small batch decoding regime. This makes decoding of memory transformers up to 5x faster than MoE models in the small batch size regime (see Figure 4).

## 4. Experimental setup

For our base model architecture, we follow closely the Llama series of dense transformers (Touvron et al., 2023; Dubey et al., 2024), which also serve as our dense baselines. We augment the base models by replacing one or more of the feed-forward layers with a shared memory layer. For scaling law experiments, we pick base model sizes of 134m, 373m, 720m, and 1.3b parameters. For these models, we use the Llama2 tokenizer with 32k tokens, and train to 1T tokens with a pretraining data mix that is similar to that of Llama2 (Touvron et al., 2023). For experiments at the 8B base model scale, we use the Llama3 (Dubey et al., 2024) configuration and tokenizer (128k tokens), and a better optimized data mix similar to Llama3.

### 4.1. Baselines

In addition to the dense baselines, we also compare to other parameter augmentations including mixture-of-experts (MOE) (Shazeer et al., 2017) and the more recent PEER (He, 2024) model. In MOE, each FFN layer is composed of multiple "experts", only a subset of which participate in the computation for each input. The PEER model is conceptually similar to a memory layer, but instead of retrieving a single value embedding, it retrieves a pair of embeddings, which combine into a rank-1 matrix. Several of these are assembled together into a dynamic feed-forward layer. PEER works similarly to memory layers in prac-

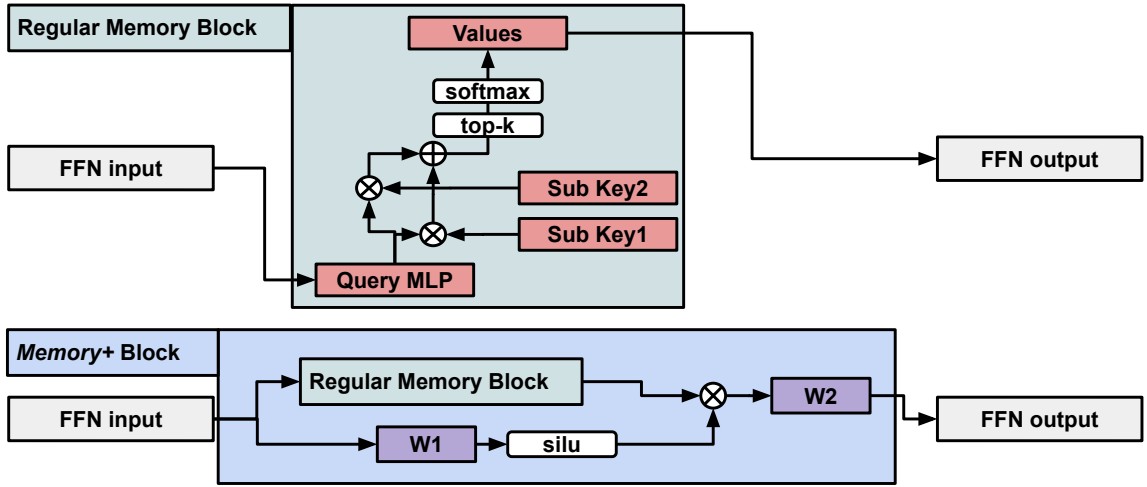

**Figure 3.** On the top, the regular memory layer. On the bottom, the `Memory+` block, with the added projection, gating and silu non-linearity

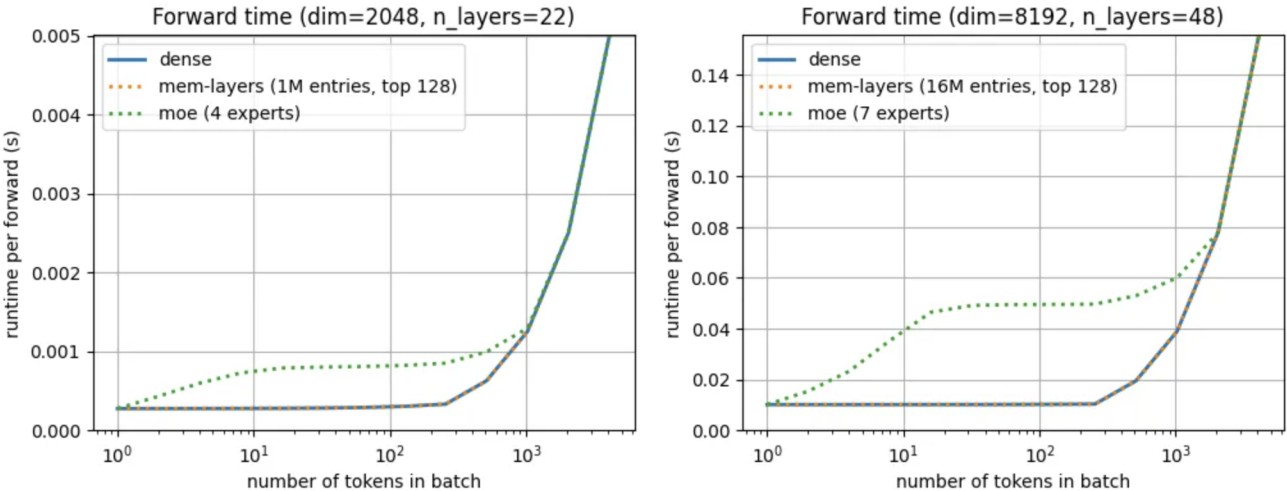

**Figure 4.** Roofline analysis of the decoding time for a dense model, an MoE model, and a Memory transformer. All models have the same number of activated parameters per token. The MoE and Memory models have the same number of total parameters.

tice, but requires twice the number of parameters for the same number of keys. Like memory layers, these methods increase the number of parameters in the model without significantly increasing FLOPs. We pick the number of experts in MOE and the number of keys in PEER to match the number of parameters of our memory-augmented models as closely as possible. MOE models are trained with expert choice (Zhou et al., 2022), and evaluated with top-1 routing. PEER layers share the same configuration and hyper-parameters as our memory layer implementation.

### 4.2. Evaluation benchmarks

Our evaluations cover factual question answering (NaturalQuestions (Kwiatkowski et al., 2019), TriviaQA (Joshi et al., 2017)), multi-hop question answering (HotpotQA (Yang et al., 2018)), scientific and common sense world knowledge (MMLU (Hendrycks et al., 2021), HellaSwag (Zellers et al., 2019), OBQA (Mihaylov et al., 2018), PIQA (Bisk et al., 2019)) and coding (HumanEval (Chen et al., 2021), MBPP (Austin et al., 2021)). We try to report the most commonly used accuracy metrics (exact match or F1 score for QA benchmarks, pass-at-1 for coding). For some bencmarks, the performance of small models can be very low, and accuracy numbers

noisy. Therefore we use negative log-likelihood (nll) of the correct answer for model ablations.

# 5. Scaling results

We compare `Memory` models to baselines in a compute-controlled setting.

## 5.1. With fixed memory size

First, we fix the size of the memory, and therefore the number of extra parameters, and compare with the dense baseline, as well as roughly parameter matched MOE and PEER models. Models with the same base model configuration have negligible differences in FLOPs. For `Memory` models, we fix the number of half keys to $2^{10}$, and thus the number of memory values to $2^{20}$ (roughly 1 million). For the PEER baseline, we pick the number of half-keys to be 768, resulting in slightly more total parameters than `Memory`. For MOE models, we pick the lowest number of experts such that the parameter count exceeds that of `Memory`. This corresponds to 16, 8, 6, and 4 experts for the 134m, 373m, 720m and 1.3b sizes respectively.

The vanilla `Memory` model has a single memory layer, which we pick to replace the middle FFN layer of the transformer. Our improved `Memory+` model has 3 memory layers, placed centered with a stride of 4 for the 134m models and 8 for the others. Additionally it includes a custom swilu non-linearity, and optimized key dimension (set to equal half of the value dim). As noted earlier, memory layers share parameters, thus have identical memory footprint to a single memory layer.

We can see from Table 1 that `Memory` models improve drastically over the dense baselines, and generally match the performance of models with twice the number of dense parameters on QA tasks. `Memory+` improves further over `Memory`, with performance falling generally between dense models with 2x-4x higher compute. The PEER architecture performs similarly to `Memory` for the same number of parameters, while lagging behind `Memory+`. MOE models underperform the memory variants by large margins. Figure 5 shows the scaling performance of `Memory`, MOE and dense models on QA tasks across various base model sizes.

## 5.2. Scaling memory size with a fixed base model

Next, we investigate scaling behaviour with respect to the memory size for a fixed base model. In Figure 1, we see that factual QA performance for a `Memory+` model keeps increasing predictably with increasing memory size. At 64 million keys (128 billion memory parameters), a 1.3b `Memory` model approaches the performance of the Llama2 7B model, that has been trained on 2x more tokens using

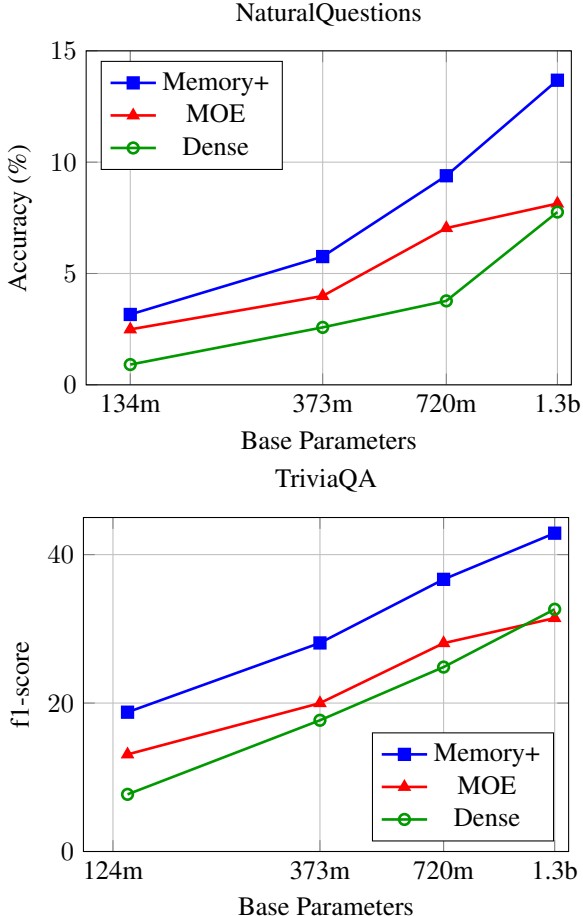

*Figure 5.* Accuracy vs. Base Parameters for NaturalQuestions and TriviaQA (Memory+ models use 1 million memory embeddings.)

10x more FLOPs. (see also Table 2).

## 5.3. Results at 8B scale

Finally, we scale our `Memory+` model with an 8B base model and $4096^2$ memory values (64B memory parameters). We use the Llama3 8B (Dubey et al., 2024) architecture and tokenizer, and train on a data mix similar to Llama3 (Dubey et al., 2024). We report results at 200 billion and 1 trillion tokens of training in Table 2. On an expanded set of benchmarks, including general scientific and world knowledge and coding, we see that memory augmented models significantly outperform dense baselines. The gains are more pronounced earlier in training (200B tokens), suggesting that memory helps models learn facts faster. At only 1 trillion tokens of training, our `Memory+` model approaches the performance of Llama3.1 8B, which was trained on 15 trillion tokens.

*Table 1.* Comparing memory augmented architectures with baseline models on QA tasks. `Memory` models have 1 million value embeddings unless otherwise specified in the model configuration column. Metrics are accuracy for NQ, PIQA, OBQA and F1 score for TQA, HotpotQA.

| Base Params | Tokens | FLOPS | Model Configuration | Total Params | NQ | TQA | PIQA | OBQA | HotPot |
|---|---|---|---|---|---|---|---|---|---|
| | | | Dense | 134m | 0.91 | 7.7 | 62.13 | 16.40 | 5.18 |
| | | | MOE | 984m | 2.49 | 13.08 | 65.78 | 18.80 | 7.80 |
| 134m | 1T | 7.9e20 | PEER | 1.037b | 2.46 | 16.34 | **67.25** | 17.40 | 8.82 |
| | | | Memory | 937m | 2.1 | 16.31 | 66.65 | **17.80** | 9.28 |
| | | | Memory+ | 937m | **3.16** | **18.77** | 65.94 | 17.60 | **9.35** |
| | | | Dense | 373m | 2.58 | 17.68 | 67.47 | 18.80 | 10.06 |
| | | | MOE | 1.827b | 3.99 | 19.94 | 68.88 | **22.20** | 12.50 |
| 373m | 1T | 2.6e21 | PEER | 1.575b | 5.1 | 26.39 | 70.19 | 21.60 | 12.96 |
| | | | Memory | 1.441b | 4.95 | 24.24 | 69.37 | 20.40 | 12.53 |
| | | | Memory+ | 1.434b | **5.76** | **28.10** | **71.22** | 22.00 | **13.34** |
| | | | Dense | 720m | 3.77 | 24.85 | 71.33 | 22.60 | 12.90 |
| | | | MOE | 2.768b | 7.04 | 28.08 | 70.08 | 20.80 | 14.10 |
| 720m | 1T | 4.9e21 | PEER | 2.517b | 7.92 | 33.26 | 71.98 | **25.00** | 14.03 |
| | | | Memory | 2.316b | 7.2 | 34.8 | 71.82 | 24.40 | **14.94** |
| | | | Memory+ | 2.316b | **9.39** | **36.67** | **72.42** | 24.00 | 14.92 |
| | | | Dense | 1.3b | 7.76 | 32.64 | 72.74 | 23.40 | 13.92 |
| | | | MOE | 3.545b | 8.14 | 31.46 | 73.72 | 25.20 | 15.15 |
| | | | PEER | 3.646b | 12.33 | 42.46 | 73.34 | 26.60 | 15.39 |
| | | | Memory | 3.377b | 9.83 | 39.47 | 72.29 | 25.80 | 15.46 |
| 1.3b | 1T | 8.5e21 | Memory+ | 3.377b | **13.68** | **42.89** | **75.35** | **26.80** | **16.72** |
| | | | Memory+ 4m | 9.823b | 14.43 | 51.18 | 75.03 | 27.80 | 18.59 |
| | | | Memory+ 16m | 35.618b | 20.14 | 58.67 | 76.39 | 26.80 | **20.65** |
| | | | Memory+ 64m | 138.748b | 20.78 | 62.14 | 77.31 | 30.00 | 20.47 |
| *llama2 7B* | 2T | 9.1e22 | Dense | 7b | 25.10 | 64.00 | 78.40 | 33.20 | 25.00 |

*Table 2.* Results with an 8B base model. `Memory+` models have 16 million memory values (64 billion extra parameters). Metrics are accuracy for NQ, PIQA, OBQA, HellaSwag, MMLU; F1 score for TQA, HotPotQA; pass@1 for HumanEval, MBPP. The number of training tokens for each model is denoted in parenthesis.

| Model (8B) | Tokens | FLOPS | HellaS. | Hotpot | HumanE. | MBPP | MMLU | NQ | OBQA | PIQA | TQA |
|---|---|---|---|---|---|---|---|---|---|---|---|
| *llama3.1* | 15T | 6.8e23 | 60.05 | 27.85 | 37.81 | 48.20 | 66.00 | 29.45 | 34.60 | 79.16 | 70.36 |
| dense | 200B | 9.1e21 | 53.99 | 20.41 | 21.34 | **30.80** | 41.35 | 18.61 | **31.40** | 78.02 | 51.74 |
| Memory+ | 200B | 9.1e21 | **54.33** | **21.75** | **23.17** | 29.40 | **50.14** | **19.36** | 30.80 | **79.11** | **57.64** |
| dense | 1T | 4.6e22 | 58.90 | 25.26 | 29.88 | **44.20** | 59.68 | 25.24 | 34.20 | **80.52** | 63.62 |
| Memory+ | 1T | 4.6e22 | **60.29** | **26.06** | **31.71** | 42.20 | **63.04** | **27.06** | 34.40 | 79.82 | **68.15** |

## 5.4. Model ablations

In this section, we present results which motivate our modelling choices for the `Memory+` architecture.

**Memory layer placement** Since the memory pool is shared, we can replace more FFN layers with memory layers without increasing either the memory or the compute budget. We see that as we add more memory layers, performance initially increases. However, as we're effectively removing dense parameters from the model for each added memory layer, eventually the model performance degrades, revealing a sweet spot at around 3 memory layers (Table 3, top). Moreover, we experiment with the placement of these layers, modifying the centring and spacing. We find that centred placements with larger strides are better, and we adopt this for our `Memory+` architecture.

**Memory layer variants** We experiment with minor modifications to the memory mechanism (Table 3, bottom). We try 1. gating the memory with the input using a linear projection, 2. adding a custom swilu non-linearity (Figure 3), 3. adding random key-value pairs in addition to the top-k during pre-training to unbias key selection, 4. adding a single fixed key-value pair (softmax sink) to the top-k selected values during pre-training to serve as "anchor". We find that the swilu non-linearity consistently improves results, and we adopt this improvement into our model. Simple gating improves performance only in some cases, and swilu already covers this behaviour to some extent, so we decide not to do additional gating. For key sampling improvements, including the random keys and the fixed (sink) key, we see minor improvements, however these have some negative impact on training speed in our implementation, and the gains were not consistent for larger model sizes, therefore we excluded them from our experiments, leaving this direction open for future exploration.

|  | nll | NQ nll | TQA nll |
|---|---|---|---|
| **layer #** | | | |
| 12 | 2.11 | 12.13 | 8.34 |
| 12,16,20 | 2.08 | 11.60 | 7.54 |
| 8,12,16 | 2.07 | 11.79 | 7.64 |
| 4,12,20 | **2.06** | **11.32** | **7.20** |
| 5,8,11,14,17,21 | 2.11 | 11.79 | 7.73 |

|  | nll | NQ nll | TQA nll |
|---|---|---|---|
| **Model** | | | |
| PK base | 2.11 | 12.13 | 8.34 |
| +gated | 2.11 | 12.24 | 8.17 |
| +swilu | 2.11 | 12.05 | 8.09 |
| +random values | 2.11 | 12.36 | 8.09 |
| +softmax sink | 2.11 | 12.19 | 8.04 |

*Table 3.* Ablation studies: on the top, number of memory layers with shared memory, on the bottom different memory architecture variations. Metrics are all log likelihood, on the training set, NQ answers and TQA answers.

**Key and value dimension** By default, the memory value dimension is chosen to be the same as the base model dimension. However, we can potentially trade-off the value dimension with the number of values in the memory without changing the total parameter size of the memory using an extra projection after `Memory`. We present this ablation in Table 4, top, and find that the default configuration is optimal. We can also independently increase the key embedding dimension, which we do in Table 4, bottom. We find unsurprisingly that increasing the key dim is beneficial. However, increasing the key dim adds more dense parame-

ters to the model, and thus we cannot increase it indefinitely without breaking fair comparisons. We pick a key dimension of half the base model dim for our experiments.

|  |  | nll | NQ nll | TQA nll |
|---|---|---|---|---|
| **v_dim** | **#values** | | | |
| 64 | 16m | 2.15 | 12.86 | 8.75 |
| 256 | 4m | 2.14 | 12.63 | 8.49 |
| 1024 | 1m | **2.11** | **12.13** | **8.34** |
| 2048 | 512k | 2.14 | 12.49 | 8.53 |

|  | nll | NQ nll | TQA nll |
|---|---|---|---|
| **key_dim** | | | |
| 256 | 2.11 | 12.13 | 8.34 |
| 512 | 2.12 | 12.32 | 8.15 |
| 1024 | 2.11 | 12.37 | 8.25 |
| 2048 | **2.09** | **11.98** | **7.83** |

*Table 4.* Ablation studies: on the top, varying the value embedding dim while keeping total parameter count the same, on the bottom varying key dim. Metrics are all log likelihood, on the training set, NQ answers and TQA answers. These were ran on the 373m model size, which uses a latent dimension of 1024. key_dim is the sum of the dimension of the sub-keys.

## 6. Implications and shortcomings of the work

Scaling of dense transformer models has dominated progress in the AI field in the last 6 years. As this scaling is nearing its physical and resource limits, it's useful to consider alternatives which might be equally scalable without being as compute and energy intensive. Memory layers with their sparse activations nicely complement dense networks, providing increased capacity for knowledge acquisition while being light on compute. They can be efficiently scaled, and provide practitioners with an attractive new direction to trade-off memory with compute.

While the memory layer implementation presented here is orders of magnitude more scalable than previous works, there still remains a substantial engineering task to make them efficient enough for large scale production uses. Dense architectures have been optimized for and co-evolved with modern GPU architectures for decades. While we believe it's in principle possible to make memory layers as fast, or even faster than regular FFN layers, we acknowledge that this needs non-trivial effort.

We have so far presented only high level empirical evidence that memory layers improve factuality of models. However, we believe the sparse updates made possible by memory layers might have deep implications to how models learn and store information. In particular, we hope that

new learning methods can be developed to push the effectiveness of these layers even further, enabling less forgetting, fewer hallucinations, and continual learning.

## Impact Statement

This goal of this work is to advance the field of ML. There are many potential societal consequences of our work, none which we feel must be specifically highlighted here.

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

## A. Training hyperparameters

Here are the hyperparameters used to train our transformer models.

*Table 5.* Model Configurations

| Model Size | Embedding Dim. | Number of Layers | Attention Heads | Learning Rate |
|---|---|---|---|---|
| 134m | 768 | 12 | 12 | $3 \times 10^{-4}$ |
| 373m | 1024 | 24 | 16 | $3 \times 10^{-4}$ |
| 720m | 1536 | 22 | 12 | $3 \times 10^{-4}$ |
| 1.3b | 2048 | 22 | 16 | $3 \times 10^{-4}$ |
| 8b | 4096 | 32 | 32 | $1 \times 10^{-4}$ |

For all model sizes, `Memory` and `Memory+` experiments use 4 heads and 32 top-k values for the memory embedding lookups. We did ablations on these values, but largely came to the same conclusions as the original product-keys paper (Lample et al., 2019). Overall, varying the number of heads or top-k while keeping their product (128) the same affects results minimally. Increasing the total keys improves little beyond 128, but has substantial memory lookup and gpu memory costs. On the other hand, decreasing to 64 has non-negligible accuracy degradation.

## B. Distribution of keys during training

We monitored the distribution of selected keys during training and noticed that the distribution tends to become more uniform as training progresses. The distributions starts heavily skewed and progressively improves as training continues.

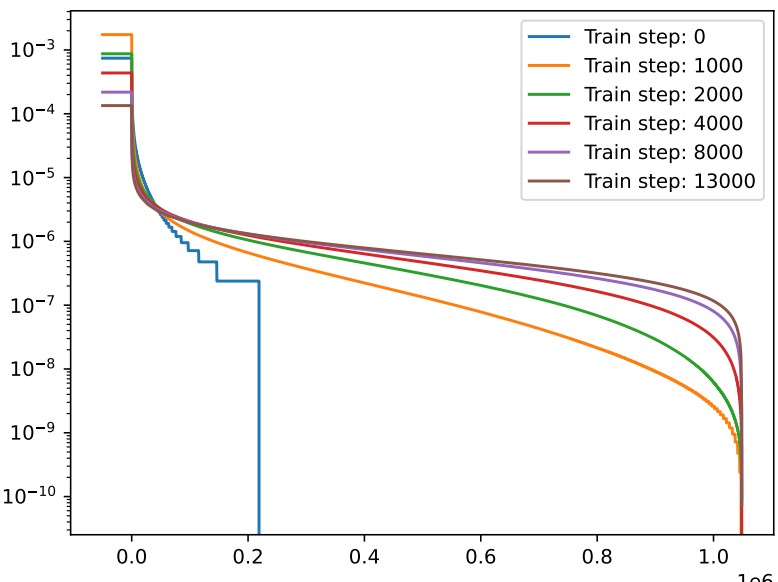

*Figure 6.* Evolution of the distribution of `Memory+` selected keys during training. Keys on the x-axis or sorted by decreasing occurrence.

