# OpenReview forum: "Memory Layers at Scale"
_ICML.cc/2025/Conference — ICML 2025 poster_

### Official Review · Reviewer_Sxwo · 2025-02-19

**Overall Recommendation:** 4

**Summary:**

This paper proposes an improved Memory layer design, which adds extra parameters to the model without increasing FLOPs. Following Memory layer design of previous works, this paper optimizes embedding bag kernels and improves architectures by gating networks for performance and QKNorm for stability.

In the experiments, the paper shows that Memory+ shows an decoding advantage than MoE where the activated parameters are relatively small under small batch size. Besides, the performance of Memory+ also outperforms Dense and MoE baselines. The model is scaled to 8B to show the effectiveness of Memory layer.

## Update after rebuttal
My score keeps as Accept. The author solved most of my concerns.

**Claims And Evidence:**

This paper makes a systematic experiment to support their claims.

**Essential References Not Discussed:**

All the essential references are discussed in my point of view.

**Experimental Designs Or Analyses:**

The experimental designs are sound in general. However, I have some concerns in the comparisons with MoE models:

1. There are many improved MoE baselines recently, for example, fine-grained and shared experts in DeepseekMoE. Usually, MoE models are at least better than Dense baselines. However, in Figure 5, the MoE models are worse than Dense in 1.3b dense. Therefore, I'm curious at the implementation details of MoE.

2. Besides the module difference, there is also an important difference, where Memory+ activate Memory layers every several layers while MoE model replace FFN with MoE in every layers. Are there any ablation studies, showing that whether layout plays a role on the performance?

**Methods And Evaluation Criteria:**

The proposed method aims at adding extra parameters to enhance model's capability without increasing FLOPs. The experiments are standard under LLM training scenarios.

**Other Comments Or Suggestions:**

Overall, this is a valuable paper and shows a promising direction for LLM sparsity. However, I have some doubts about the details in the experiments. I list them in different sections of this review. I hope a useful discussion can help us better understand and position this paper.

**Other Strengths And Weaknesses:**

Strength:
1. I think this network design will make an influence on hardware design. The computation patterns are different from current dense matrix multiplications.
2. LLM needs to go beyond current MoE designs for further sparse ratio.

Weaknesses:
1. Following Experimental Designs Or Analyses, I think there should be more detailed ablations and comparisons with MoE and layout design.

**Questions For Authors:**

1. I'm not fully convinced that the decoding efficiency of Memory+ layer is consistent with dense baseline across all the batch size. When the batch size is increasing, the activated memory ratio will increase, where the memory access demand also increases. Compared with a linear projection, how can the curve almost the same?

2. For model design philosophy, the same sub key2 is used among different sub key1, which means there are some shared knowledge in the same row or column. Is there any explanation for that?

3. Is there any experiment about the relation between performance and sparsity ratio? In other words, when the sparsity ratio is further increasing, can the performance also improve consistently?

**Relation To Broader Scientific Literature:**

This paper is an improved work of Memory network, which is a kind of sparse activation modules. Besides, it also has connections with Mixture-of-Experts.

**Theoretical Claims:**

No Theoretical claims.

---

> ### Author Rebuttal · Authors · 2025-04-01
>
> Thank you for the kind review! We will try to address the comments in the order they were raised.
>
> ## MoE Results
> While we cannot fully explain the poor performance of MOE models (especially for the 1.3b setting), we can provide the following additional information:
> - Our MOE implementation reuses code from a recent SoTA MOE architecture paper (reference not given to protect anonymity), which we believe to be well optimized for MOE.
> - We have independent evidence from other internal teams that match our MOE results
> It is quite surprising that MoE performs slightly worse than the dense model for NQ and TQA on Figure 5. MoE does perform better than dense at the 1.3b scale on all other tasks of Table 2, PIQA, ObQA and HotPot.
>
> ## Memory+ layout
> > Are there any ablation studies, showing that whether layout plays a role on the performance?
>
> Table 3 (top) provides some insights into how the memory layers layout (number of layers, centering and spacing) affects performance. In the table, adding more Memory+ layers beyond 3 decreases performance. We believe this is because for every memory layer we add, we remove the FFN layer, resulting in effectively less computation.  Also, we share the value parameters across all the Memory+ layers of the model, thus the total parameter count does not increase.  As a result, there is a sweet spot after which adding more memory layers doesn’t help.  It is possible to add memory layers in addition to or parallel to an FFN layer, however we did not consider this as it would break our compute-controlled comparisons.
>
> ## Decoding Efficiency of Memory+
> > I'm not fully convinced that the decoding efficiency of Memory+ layer is consistent with dense baseline across all the batch size. When the batch size is increasing, the activated memory ratio will increase, where the memory access demand also increases.
>
> You are indeed correct that Memory+ layers memory access grows with batch size.  However, for the top-k value that we pick (4 heads, 32 top-k, 128 total), the memory demand remains comparable to the dense model for small batch sizes (note that the analysis for one forward pass through the entire model, not just a single memory layer).  For large batch sizes, all models are compute bound, thus the curve follows the dense models curve closely for the full range of batch sizes.  However, MOE models quickly need a lot more memory bandwidth even at small batch sizes, and are thus severely memory bound in this regime.
>
> ## Model Design Philosophy
> Our design is built on top of the work of Large Memory Layers with Product Keys (https://arxiv.org/abs/1907.05242) in which they use two independent sets of sub keys that they then combine into a complete key.  This is done for efficiency reasons, since otherwise it would be infeasible to compute top-k over millions of keys.  This likely causes some degradation in key lookup (similar to product quantization).   On the other hand, the value parameters are kept in a flat table and are independent of each other.
>
> ## Relation between performance and sparsity ratio
> > Is there any experiment about the relation between performance and sparsity ratio? In other words, when the sparsity ratio is further increasing, can the performance also improve consistently?
>
> In Table 2 at the 1.3b scale, we study the relation between increasing the Memory+ parameter count (which increases sparsity) and performance. At the scales studied, it seems increasing the parameter count increases performance. However, we were unable to scale further in this research work and could not verify if further increase always results in better performance.
>
> Another way to increase sparsity would be to decrease k.  We find that doing this (e.g. go from top-128 to 64) hurts performance.  We also did ablations on these values, but largely came to the same conclusions as the original product-keys (Lample et al) paper, so decided not to repeat those results here.  Overall, varying heads or k while keeping their product the same effects results minimally.  Increasing the total keys improves little beyond 128, while having substantial memory lookup and gpu memory costs, while decreasing to 64 has non-negligible accuracy degradation.  In the end we stick to 128, which was also the default for the original paper.

---

> > ### Comment · Reviewer_Sxwo · 2025-04-02
> >
> > Thanks for your reply. Your explanation is comprehensive and solid, which solves most of my questions.

---

### Official Review · Reviewer_9R67 · 2025-03-14

**Overall Recommendation:** 4

**Summary:**

The paper proposes a trainable key-value look-up to embed within the transformer architecture as an inductive bias for memory. To make this computationally efficient, they propose ways to parallelize the search across GPUs and show that this architectural change helps factuality.

### Update after rebuttal
While there remains questions on generalization to unseen or new domains at inference-time and how the memory layer could actually be detrimental, I believe the paper proposes a novel architecture that acts as an inductive bias for trainable memory and demonstrate that it reliably improves performance on especially factuality-based tasks. I believe this paper should be accepted and maintain my score of 4.

**Claims And Evidence:**

Yes, the claims seem to be sound and well supported.

**Essential References Not Discussed:**

NA

**Experimental Designs Or Analyses:**

The experimental designs seem to be sound.

**Methods And Evaluation Criteria:**

Yes, the methods are sound and the evaluation across different QA benchmarks is comprehensive.

**Other Comments Or Suggestions:**

None

**Other Strengths And Weaknesses:**

Strengths
- The paper is very clearly written and the contributions of the paper are very clear. The authors show that they can in general improve the performance for factuality-related tasks upon other model architectures or training paradigms with similar or larger parameter count. Also, they show that they can maintain the runtime efficiency of dense models (over MoEs).
- Evaluations are comprehensive across different QA benchmarks.

Weaknesses
- Although the results on the knowledge-based benchmarks are comprehensive and convincing, the generality of the finding across different types of tasks like math or reasoning benchmarks hasn't been tested and seems important given that they are proposing an architecture-level change for the future generation of models to explore.

**Questions For Authors:**

One of the obvious weakness of embedding these trainable memory layers, in lieu of retrieval-based systems like RAG, is that it is difficult to swap knowledge at inference-time (although RAG can also be attached on top of this model). But, because of this inductive bias of the memory layers serving as "memory retrieval" within the model, I am curious how this may affect out-of-distribution tasks or reasoning in new domains unseen during pretraining.

**Relation To Broader Scientific Literature:**

The contributions of the paper are related to general machine learning and specifically in the field of investigating newer and different architectures that separate memory from reasoning, such as Neural Turing Machines.

**Theoretical Claims:**

There are no substantial theoretical claims in this paper.

---

> ### Author Rebuttal · Authors · 2025-04-01
>
> We thank the reviewer for the supportive feedback!  For benchmarks beyond factuality, we tried to include a variety of standard benchmarks on table 2 for the 8B models.  In addition to this, here are results for the GSM8K math benchmark that we recently ran (8B, 1 trillion tokens):
> | GSM8K exact match |  |
> | --- | --- |
> |  dense    |       35.8  |
> | Memory+  |    43.4  |
>
> > I am curious how this may affect out-of-distribution tasks or reasoning in new domains unseen during pretraining
>
> While this is a valid and interesting question, at this time we do not have any data to help us speculate on this issue.  However, we are continuing research with memory models, including in new domains of knowledge, and hope to contribute in this direction in future work.

---

### Official Review · Reviewer_5Zym · 2025-03-14

**Overall Recommendation:** 3

**Summary:**

This paper describes a scaling analysis of memory layers and a comparison with alternative sparsely activated layers like MoEs and PEER. The main claims that the paper makes are:
1. Performance improves by increasing the size of the memory layers.
2. Memory layers significantly outperform dense layers.
3. Memory layers outperform mixture-of-experts architectures with matching compute and parameter size.
4. Memory layers are faster than mixture of expert layers at low batch sizes during decoding.

There are several additional claims made via ablation studies:
5. Replacing more than three FFN layers degrades performance.
6. The Memory+ block with added projection, gating and silu improves performance

**Claims And Evidence:**

1. This claim is nicely supported by the results in Figure 1. However, it would be helpful to include a similar plot for non-factual tasks (e.g. HumanEval or MMLU).
2. This claim is well supported.
3. I some have concerns with the quality of evidence for this claim. The results in Table 1 appear to be a memory-matched comparison, but I can’t find a compute-matched comparison anywhere in the paper. Also, it looks like a comparison was made only for a single hyperparameter configuration.  How would varying the number of experts and size of experts affect the performance relative to memory layers.
4. The evidence for this claim is in Figure 4. A limitation is that the MoE baselines are implemented with very low expert counts. Would there still be a significant speedup, if more experts were used in the MoE layer?
5. This result is well-supported.
6. This result is well-supported.

**Essential References Not Discussed:**

None that I am aware of.

**Experimental Designs Or Analyses:**

See claims and evidence.

**Methods And Evaluation Criteria:**

See claims and evidence.

**Other Comments Or Suggestions:**

I am quite interested in the relationship between MoE and memory layers. For certain choices of hyperparameters, they appear to be basically equivalent. So, my main concern with the presentation and experiments in this paper center around the categorical nature of the MoE claims (*i.e.* that MoEs are worse than Memory layers).

I think the the paper would be much strengthened by carefully analyzing these hyperparameters and understanding how these techniques relate.

Typos and nits:

- In Figure 1, the caption should read top/bottom not left/right.

Typos and nits:
- In Figure 1, the caption should read top/bottom not left/right.

**Other Strengths And Weaknesses:**

There are several important open questions, which are not addressed in the paper.
- How does the efficiency of a dense model compare to the memory model at training and inference?
 - What is the value of k chosen and how does it affect performance?

**Questions For Authors:**

No questions.

**Relation To Broader Scientific Literature:**

This work provides a scaling analysis of a well-known technique from the literature. Section

**Theoretical Claims:**

N/A

---

> ### Author Rebuttal · Authors · 2025-04-01
>
> We thank the reviewer for their thoughtful review.  Please find our answers and clarifications below:
>
> ## Evidence for claims
> > it would be helpful to include a similar plot for non-factual tasks (e.g. HumanEval or MMLU)
>
> We opted to not provide numbers for these benchmarks for the small model sizes, since they are often not meaningful (e.g. MMLU is ~25% which is chance level).  However, here are the nll values for the two mentioned benchmarks at 1.3b scale for reference (lower is better):
> | Model    | HumanEval nll    | MMLU nll |
> | ---------- |   -------------------- | -------- |
> | dense   |          53.52          |  1.23  |
> | MOE     |          52.29         |  1.17  |
> | Mem+   |          52.02          |  1.14  |
>
> > I can’t find a compute-matched comparison anywhere in the paper
>
> All of our experiments are compute (FLOPs) matched within the same scale category.  E.g. for the 1.3b scale, the dense, MOE and memory layer models all have (almost) identical flop cost.  For the MOE-memory layer comparisons, we also attempted to match parameter count.  We will make it more clear in the paper that the comparisons are FLOP-controlled, as this is one of the main claims of the paper.
>
> > How would varying the number of experts and size of experts affect the performance relative to memory layers
>
> For the main experiments, we set the number of memory values at 2^20 for the memory models, and tried to set the MOE experts to match the parameter count.  This choice results in varying number of MOE experts for different scales (e.g. 16 experts for the 134m model, 4 experts for the 1.3b model).  While we agree that having more experiments at various parameter / expert counts would be ideal, we did not have the needed resources to run them.  We would guess that having more but smaller experts would behave more similarly to a memory layer.  In fact, the PEER work which we compare against, investigates the limit of this, where they have up to a million rank-1 experts.  Our work performs comparably or better, while being simpler and more compute efficient.
>
> ## Other questions and comments
> > How does the efficiency of a dense model compare to the memory model at training and inference?
>
> We provide such an analysis in figure 4, plotting latency against batch size for dense, MOE and memory models.  Memory models have similar efficiency with dense models in both the small batch-size (typical inference) and large batch-size (typical training) regimes, while MOE inference latency is much higher.  This is a roofline analysis.  In practice, we observe ~10% lower training throughput compared to a dense model during training at 8B scale, due to communication overheads.
>
> > What is the value of k chosen and how does it affect performance?
> Thank you for catching this oversight!  We used 4 heads and k=32 (total keys 128) for all experiments, we will add this to the paper.  We also did ablations on these values, but largely came to the same conclusions as the original product-keys (Lample et al) paper, so decided not to repeat those results here.  Overall, varying heads or k while keeping their product the same effects results minimally.  Increasing the total keys improves little beyond 128, while having substantial memory lookup and gpu memory costs, while decreasing to 64 has non-negligible accuracy degradation.  In the end we stick to 128, which was also the default for the original paper.
>
> ## Typos and nits
> Thank you for pointing these out, we will edit this in the final version of the paper.

---

### Official Review · Reviewer_orhQ · 2025-03-14

**Overall Recommendation:** 4

**Summary:**

This paper proposes to replace the feed-forward layer in LLM with a memory layer. A memory layer consists of a key and a value matrix. Similar to the attention mechanism, each token representation will attend to the top-k selected values. Since it's sparsely activated, the computation cost will be much lower than the original feed-forward layer, and is more merory-bound.

Through dedicated engineering design, the authors can successfully scale up the memory layer up to 128B parameters for LLMs in the size of 134M-8B. Compared to baselines (dense LLM and MoE), LLM with memory layer performs significantly better across various tasks, while requiring less compute than the dense LLM and comparable compute to MoE.

**Claims And Evidence:**

Yes, all claims are well supported.

**Essential References Not Discussed:**

No, the paper thotoughly discuss all closely related works.

**Experimental Designs Or Analyses:**

Yes, the experimental designs are very sound and valid.

**Methods And Evaluation Criteria:**

Yes, all evaluation criteria make sense for me, with enough experimental results to support the claims.

**Other Comments Or Suggestions:**

None

**Other Strengths And Weaknesses:**

Strengths:
1. The paper is well-written.
2. The experimental results are thorough and promising, clearly demonstrating the effectiveness of memory layer.


Weaknesses:
1. Missing experimental details. The experimental setup is not very detailed, most training hyper-parameters are not shown.
2. Since this is an engineering paper, and the authors do a lot contributions to the optimization side (like kernel), it would be better to include such materials for review.

**Questions For Authors:**

1. I would like to review the implementation of the custom EmbeddingBag, since this is the key contribution for the speed up. But there is little implementation detail in the paper. I'm willing to raise my score, if the authors can show the implementation and explain it in details for my validation, because it's the key for memory layer at scale.
2. Do you observe any unbalance unsage of the key and value? Since the key and value matrices are in large N, I wonder whether such a memory layer will have unbalance problem, i.e. only a limited amount of keys and values are used. It would be good to show the usage rate of the key and values.

**Relation To Broader Scientific Literature:**

This paper is related to efficient training and inference of LLM, and offers very promising results. It is a scale-up work of previous memory layer [1], also making it more efficient.


[1] Sukhbaatar, S., szlam, a., Weston, J., and Fergus,R. End-to-end memory networks.

**Theoretical Claims:**

This is an engineering paper without theoretical proof.

---

> ### Author Rebuttal · Authors · 2025-04-01
>
> Thank you so much for the positive feedback! We will do our best to improve based on the reviews. Here is how we plan on addressing the comments:
>
> ## Experimental details
> We will add an appendix with the details about the experimental setup including model dimensions and training hyper-parameters. Here are some of these details.
> For 134m base models, we use dim of 768, 12 layers and attention with 12 heads.
> For 373m base models, we use dim of 1024, 24 layers and attention with 16 heads.
> For 720m base models, we use dim of 1536, 22 layers and attention with 12 heads.
> For 1.3b base models, we use dim of 2048, 22 layers and attention with 16 heads.
> For 8b base models, we use dim of 4096, 32 layers and attention with 32 heads.
> In all Memory and Memory+ experiments, we use 4 heads and 32 top-k values.
> We used a learning rate of 3e-4 for the 134m to 1.3b models and 1e-4 for the 8B models.
>
> ## Code release for EmbeddingBag
> Our code is already open source (withholding reference due to anonymity), however here is the relevant file with the kernel implementation for your review: https://justpaste.it/cb6xz .
>
> ## Unbalanced key usage
> Regarding unbalanced usage of keys, we notice that there can be a few steps at the very beginning of training during learning rate warmup where the keys become quite unbalanced. This issue goes away on its own and the distribution naturally smooths out during training.  In general, we did not need to do any regularization to enforce balance.  We will add a plot showing how this distribution evolves during the first few steps of training in the appendix.

---

### Official Review · Reviewer_3YhT · 2025-03-18

**Overall Recommendation:** 4

**Summary:**

The authors conduct LLM scaling experiments in which the dense FFNs in a transformer are replaced with "Memory Layers".  A Memory Layer uses the attention operation to attend over a block of trainable parameters.  The advantage of attention over a traditional MLP, is that it is possible to implement sparse variations of attention; in this case the authors use product-quantized keys (Lample et al., 2019) to implement a sparse Top-K lookup.

Top-K attention with memory layers is similar in concept to sparse mixture-of-experts (MoE) architectures.  In both cases, the number of trainable parameters in the FFN can be dramatically increased, because those parameters are now sparsely activated, thus allowing more parameters to be utilized without a corresponding increase in FLOPs.

The authors compare Top-K Memory Layers against standard (dense) transformers, sparse MoE models, and the newer PEER architecture, which similarly uses product-quantized keys.

**Claims And Evidence:**

I am surprised by the poor performance of the MoE models in Figure 5.  MoE models are quite similar in many ways to product keys, where the first "key" of the pair determines the choice of expert.  Thus, for equal numbers of parameters, I would expect the MoE model to be very close to Memory, perhaps lagging slightly behind. Why is there such a big difference between them?  And why does the performance of MoE drop so sharply at the 1.3b parameter mark?  That seems odd.

In fact, looking at Table 2, MoE and Memory may be relatively close, it's just that Figure 2 shows Memory+ rather than Memory.  Perhaps you could put curves for both Memory and Memory+ into Figure 2?

Table 1 should also specify the dimension of the values stored in memory.  If there are $2^{20}$ values, then I'm a bit confused about why there are only 984m total parameters.  In general, full architectural details (number of layers, embedding size, etc.) should at least be in the appendix.

**Essential References Not Discussed:**

Gupta et. al., "Memory-efficient transformers via top-k attention," also uses a top-k lookup mechanism.
Sukhbaatar et. al.  "Augmenting Self-attention with Persistent Memory" replaces MLPs with dense attention over trainable keys and values.

**Experimental Designs Or Analyses:**

The experiments are well designed, and properly attempt to compare architectures by balancing number of parameters.

**Methods And Evaluation Criteria:**

In general, the experiments seem to have been well designed, and make fair comparisons between architectures.

The biggest weakness of this paper is that the discussion of performance with respect to implementation is lacking, especially regarding parallelism and batching.  When training a model with a sequence length of 8k, which is common, there are batch_size x 8k query vectors that must be processed in parallel.  Both dense MLPs and MoEs can easily group the queries into large batches.  The large batch size means that attention can be done with a matrix-matrix multiply, and the keys and values can be read from memory in blocks, which is very efficient.

In contrast, a sparse Top-K algorithm must select a *different* set of values for each query vector, which requires a sparse read from memory.

As a result, I would expect Top-K lookup to be very fast for inference, when the batch size is small, because the values are sparsely activated.  (The authors do in fact make this claim.)  However, I would expect it to be significantly slower during training, when large numbers of queries have to be processed in parallel.  I would have liked to see a chart that compares the lookup speed of Dense vs. MoE vs. Top-K for different sized batches of query vectors.  The authors do mention some scaling challenges in Section 6, but do not give details.

**Other Comments Or Suggestions:**

The term "Memory" is rather overused at this point, and can mean many different things.  Instead of using the word "Memory Layer", I would use the term "Top-K Memory Layer", especially in the title, abstract, and introduction.

**Other Strengths And Weaknesses:**

N/A

**Questions For Authors:**

See above.

**Relation To Broader Scientific Literature:**

The authors cite the appropriate literature.

**Theoretical Claims:**

It should be noted that, contrary to the claims in (Lample et al., 2019), product keys are an approximate Top-k lookup algorithm; it is not guaranteed to return the actual Top-k keys.  To see why, assume that A,B,... are half-keys of dimension d/2, which have been normalized to unit length.  Assume we are searching for the key AB, and the memory contains the following:

* 500 keys of the form AX, where X$\cdot$B $\leq$ 0.
* 500 keys of the form YB, where Y$\cdot$A $\leq$ 0.
* A'B' where A$\cdot$A' = 0.9 and B$\cdot$B' = 0.9.

The closest matching key to the query for AB is A'B' (AB $\cdot$ A'B' = 1.8).
However, with K=128, the product key algorithm will fail to find it.

Please update the text to mention that product keys are an approximate Top-k algorithm.

---

> ### Author Rebuttal · Authors · 2025-04-01
>
> Thank you for your detailed review and thoughtful comments!  We will try to address the feedback in order of priority, starting with the discussion of performance as this was deemed to be the “biggest weakness” of the paper.
>
> ## Discussion of performance
> > The biggest weakness of this paper is that the discussion of performance with respect to implementation is lacking, especially regarding parallelism and batching
>
> The reviewer’s intuition and observations in this section about the performance of memory layers vs. MOE or dense models is correct. Memory layer models have a large advantage compared to MOE at inference time, where decoding batch sizes are small.  At large batch sizes, the three models are in principle equivalent.  This is the typical setting for training.  In practice, memory layers do have some additional communication overhead, however we have been able to realize training throughput that is within 10% of the dense model training speed using our optimized implementation.
>
> > I would have liked to see a chart that compares the lookup speed of Dense vs. MoE vs. Top-K for different sized batches of query vectors
>
> We provide such a chart in figure 4, where we plot the decoding time (dominated by memory lookup for small batch sizes) of the 3 architectures against the batch size.
>
> ## MOE results
>  > I am surprised by the poor performance of the MoE models
>
> While we cannot fully explain the poor performance of MOE models (especially for the 1.3b setting), we can provide the following additional information:
> - Our MOE implementation reuses code from a recent SoTA MOE architecture paper (reference not given to protect anonymity), which we believe to be well optimized for MOE.
> - We have independent evidence from other internal teams that match our MOE results
>
> That being said, the 1.3b MOE results do not fit the scaling trends.  We don’t know the cause of this, but it might be because the MOE configuration which is parameter matched to memory layer at this scale only has 4 experts, which is quite small.
>
> > If there are 2^20 values, then I'm a bit confused about why there are only 984m total parameters
>
> The value dimension in the memory layer is equal to the model dimension, which is different for each dense model scale.  For the 134m model, this is 768.  This is indeed confusing, and we will provide a table of model parameters for each scale in the appendix.
>
> ## Theoretical claims
> The reviewer is mostly correct, however the way product keys are implemented, we have 2 sets of half-keys, and the full set of keys is defined as the product of these sets.  As a result, if A is in the first set and B is in the second set, then AB is by definition part of the key set.  For the example given by the reviewer, if A’B’, AX and BY are in the keys, then so are A’B, AB’, which both have higher scores than A’B’.
>
> ## Additional references
> Thank you for providing additional references, we will include them in the final version.

---

> > ### Comment · Reviewer_3YhT · 2025-04-03
> >
> > Thank you for the clarification about theory -- that makes sense.  :-)
> >
> > Hmmm.  I figured that I must not have been interpreting figure 4 correctly.  If your implementation can do Top-K lookup at large batch sizes with only 10% overhead, then that is surprising to me, and I'm impressed.

---

### Decision · Program_Chairs · 2025-05-01

**Decision:**

Accept (poster)

**Comment:**

This paper investigates Memory Layers—a sparse, trainable key-value mechanism that replaces standard feedforward layers in transformers—and introduces a refined variant called Memory+, which incorporates projection, gating, and normalization. Through a comprehensive scaling study (from 134M to 8B parameters trained on 1T tokens), the paper demonstrates that Memory Layers scale effectively and outperform both dense and Mixture-of-Experts (MoE) baselines on a range of factual QA and reasoning benchmarks. The authors also present decoding latency and efficiency comparisons using roofline analysis.

The reviewers generally praised the paper for its architectural clarity and thorough experimental validation, highlighting several strengths:

- Memory Layers provide sparse computation with lower decoding latency than MoEs, particularly in low-batch inference scenarios.

- Compared to dense models, Memory Layers achieve better accuracy under equal-FLOP budgets.

- The empirical study includes comparisons across dense and MoE baselines, layer placement ablations, factual QA performance, and some reasoning tasks.

Meanwhile, the paper contains a number of limitations that were noted by some reviewers but not treated as major concerns, and which, in my view, deserve greater emphasis:

- The highlighted benefit of inference efficiency over MoEs is demonstrated only through a roofline analysis, without a practical implementation or end-to-end latency measurement.

- There is no discussion of training efficiency for Memory Layers compared to MoEs or dense models. As noted by Reviewer 3YhT, both dense MLPs and MoEs can group queries into large batches, enabling efficient memory access and compute via block-based operations. In contrast, Memory Layers introduce fine-grained, dynamic sparsity that is not well-supported on standard accelerators. This type of sparsity is usually leveraged for gaining efficiency during inference instead of training (e.g., [a]).

- While Memory Layers outperform dense models under equal-FLOP comparisons, the paper does not examine how FLOP reductions translate into wall-clock benefits. This is a critical omission when claiming practical efficiency improvements.

- Much of the claimed benefit of Memory Layers derives from activation sparsity, but the paper omits discussion of highly relevant prior work showing that intermediate activations in transformers are either naturally sparse (e.g., [b]) or can be made sparse through architectural or training modifications (e.g., [c]). These works imply that FLOP-based comparisons may be unfairly biased against dense models, as the actual FLOP count for dense models with sparse activations is often significantly lower than nominal estimates.

[a] PowerInfer: Fast Large Language Model Serving with a Consumer-grade GPU, SOSP 2024
[b] The Lazy Neuron Phenomenon: On Emergence of Activation Sparsity in Transformers, ICLR 2023
[c] ProSparse: Introducing and Enhancing Intrinsic Activation Sparsity within Large Language Models, 2024

**Recommendation**: Acceptance. This paper introduces Memory Layers, demonstrating through a comprehensive scaling study and strong empirical results that this novel sparse architecture outperforms dense models at iso-FLOPs and shows advantages over MoEs on key benchmarks. While limitations regarding practical latency data, training efficiency details, and full context within sparsity literature exist, these are areas for refinement. The core contribution—validating a viable, scalable, and high-performing alternative architecture—warrants acceptance, expecting authors will address limitations in the final version.